# Framework for Participatory Quantitative Health Impact Assessment in Low- and Middle-Income Countries

**DOI:** 10.3390/ijerph17207688

**Published:** 2020-10-21

**Authors:** Meelan Thondoo, Daniel H. De Vries, David Rojas-Rueda, Yashila D. Ramkalam, Ersilia Verlinghieri, Joyeeta Gupta, Mark J. Nieuwenhuijsen

**Affiliations:** 1Centre for Research in Environmental Epidemiology (CREAL), Barcelona Institute for Global Health (ISGlobal), 08003 Barcelona, Spain; 2Amsterdam Institute for Social Science Research (AISSR), University of Amsterdam, 1018 WV Amsterdam, The Netherlands; d.h.devries@uva.nl (D.H.D.V.); j.gupta@uva.nl (J.G.); 3Faculty of Medicine and Health Sciences, University of Barcelona (UB), 08036 Barcelona, Spain; 4Department of Environmental and Radiological Health Sciences, Colorado State University, Fort Collins, CO 80523, USA; david.rojas@colostate.edu; 5Faculty of Social Sciences, University of Mauritius, Reduit 80837, Mauritius; yashila.ramkalam@gmail.com; 6Transport Studies Unit, University of Oxford, Oxford OX1 3QY, UK; ersilia.verlinghieri@ouce.ox.ac.uk; 7Active Travel Academy, University of Westminster, London W1B 2UW, UK; 8Department of Biomedicine, University Pompeu Fabra (UPF), 08005 Barcelona, Spain; 9Department of Environmental Epidemiology, Municipal Institute of Medical Research (IMIM-Hospital del Mar), 08003 Barcelona, Spain; 10Department of Epidemiology and Public Health, CIBER Epidemiología y Salud Pública (CIBERESP), 28029 Madrid, Spain

**Keywords:** health impact assessment, participatory approaches, evidence-base policy making, developing countries, governance

## Abstract

Background: Conducting health impact assessments (HIAs) is a growing practice in various organizations and countries, yet scholarly interest in HIAs has primarily focused on the synergies between exposure and health outcomes. This limits our understanding of what factors influence HIAs and the uptake of their outcomes. This paper presents a framework for conducting participatory quantitative HIA (PQHIA) in low- and middle-income countries (LMICs), including integrating the outcomes back into society after an HIA is conducted. The study responds to the question: what are the different components of a participatory quantitative model that can influence HIA implementation in LMICs? Methods: To build the framework, we used a case study from a PQHIA fieldwork model developed in Port Louis (Mauritius). To explore thinking on the participatory components of the framework, we extract and analyze data from ethnographic material including fieldnotes, interviews, focus group discussions and feedback exercises with 14 stakeholders from the same case study. We confirm the validity of the ethnographic data using five quality criteria: credibility, transferability, dependability, confirmability, and authenticity. We build the PQHIA framework connecting the main HIA steps with factors influencing HIAs. Results: The final framework depicts the five standard HIA stages and summarizes participatory activities and outcomes. It also reflects key factors influencing PQHIA practice and uptake of HIA outcomes: costs for participation, HIA knowledge and interest of stakeholders, social responsibility of policymakers, existing policies, data availability, citizen participation, multi-level stakeholder engagement and multisectoral coordination. The framework suggests that factors necessary to complete a participatory HIA are the same needed to re-integrate HIA results back into the society. There are three different areas that can act as facilitators to PQHIAs: good governance, evidence-based policy making, and access to resources. Conclusions: The framework has several implications for research and practice. It underlines the importance of applying participatory approaches critically while providing a blueprint for methods to engage local stakeholders. Participatory approaches in quantitative HIAs are complex and demand a nuanced understanding of the context. Therefore, the political and cultural contexts in which HIA is conducted will define how the framework is applied. Finally, the framework underlines that participation in HIA does not need to be expensive or time consuming for the assessor or the participant. Yet, participatory quantitative models need to be contextually developed and integrated if they are to provide health benefits and be beneficial for the participants. This integration can be facilitated by investing in opportunities that fuel good governance and evidence-based policy making.

## 1. Introduction

Health impact assessments (HIAs) aim to enable policymakers and other stakeholders to assess the impacts of their decisions primarily before but also after enforcing policies, projects or programs. HIA is defined as a mixed-method process to systematically judge the potential effects that a proposed intervention might have on health and the distribution of those health effects within a population [1]. HIAs are more likely to influence decision-making when the process encourages participation of decision-makers [2], local communities [3,4] and vulnerable groups [5].

There has been a growing interest in using participatory approaches in impact assessments for improving health in local communities and cities. The use of participation in health care policy was underlined in the 1978 Alma-Ata Declaration IV of the World Health Organization (WHO) [6]: ‘The people have the right and the duty to participate individually and collectively in the planning and implementation of their health care’. However, participatory HIAs are still more an exception rather than a rule [7], even in countries with more experience in HIA practice, such as Australia [8,9], Italy [10], the USA [7] and the Netherlands [11].

There is consensus that participation needs to be inherent to the HIA process [1,9,12], even though it has both benefits and challenges [13,14]. Participatory HIAs can contribute to scoping and prioritizing impacts, ensuring the objectivity and suitability of findings, and reducing costs triggered by potential public objections, providing an opportunity to value local knowledge, and improving relations between local communities and agencies [15,16,17]. Additionally, participation is generally more necessary in moderately and unstructured problems where there is less consensus on science and values [18]. Participatory processes move the costs of addressing trade-offs to the ex-ante stage rather than the ex-post stage.

Participatory methods can be applied in both qualitative and quantitative HIAs. Participation is more commonly applied in qualitative HIAs [19]. Few quantitative HIAs are participatory [20], even though quantitative HIAs are primarily designed to provide estimates that stakeholders can use for evidence-based policy making [21,22]. Hereafter, we refer to such HIAs as participatory quantitative HIAs (PQHIAs). In addition to the lack of PQHIAs, very little is known on ‘how’ participatory HIAs are currently conducted [15] and what resources are needed to do so. Few studies focus on the costs of participation, what is counted, who pays, and whether projects adequately budget for participation [23].

This study aims to fill these gaps by focusing specifically on participation in quantitative HIAs, in the sector of urban transport planning and in the context of low- and middle-income countries (LMICs). It builds on previous work addressing the lack of participatory quantitative HIA models to assess the overall burden of mortality and morbidity related to urban transport planning and the implication of HIA practice for stakeholders to establish policies in favor of healthy and environmentally sustainable cities [19,20,24,25,26]. It addresses the research question: what are the different components of a participatory quantitative model and how can these influence HIA implementation? Two sub-questions are addressed: (1) what resources are needed to conduct PQHIA in an LMIC setting? (2) What factors are likely to influence PQHIAs and the uptake of their outcomes? Our starting point is that PQHIA can be applied to different urban contexts to assess the health consequences of different environmental and lifestyle determinants within policy scenarios. Our PQHIA framework was developed based on fieldwork in Mauritius and then discussed for applicability in cities of LMICs.

We use the terms ‘participatory approaches’, ‘participation’ and ‘stakeholder engagement’ interchangeably. Such interactions can go from one-way collaboration in decision-making to empowered action triggered by individuals, informal groups or within formal partnerships [2]. This definition is wider than ‘citizen participation’ [27]. WHO states that: ‘participatory HIA consists of the ‘identification of a large number of relevant people, groups and organizations […] and the implication of stakeholders in a meaningful way, allowing their messages to be heard’ [28]. We operationalize this definition by considering (1) stakeholders as laypersons, health practitioners and policymakers most likely to use HIA in their fields, (2) sampling of participants based on their ability to act on HIA outcomes and (3) organizing recurrent meetings between the researcher and the stakeholders (see Section 2.1).

## 2. Methods

To build the framework, we used an existing PQHIA fieldwork model (see Figure 1) as a baseline and tested it in a HIA case study (Appendix A) on urban transport planning in Port Louis in Mauritius [24]. We have modified the model based on data extracted from ethnographic fieldnotes, individual interviews (IDIs), focus group discussions (FGDs) and feedback exercises with 14 stakeholders. We followed qualitative thematic analysis methods to analyze the data and structure study findings [29]. We confirmed the validity of the ethnographic data using five quality criteria: credibility, transferability, dependability, confirmability, and authenticity. All subjects gave their informed consent for inclusion before they participated in the study. The study was conducted in accordance with the Declaration of Helsinki, and the protocol was approved by the National Ethics Committee of the Ministry of Health and Quality of Life in Mauritius (project protocol MHC/CT/NETH/THONM) and by the Ethical Advisory Board of the Amsterdam Institute for Social Science Research (AISSR).

### 2.1. Case Study

Mauritius is a small island developing state (SID) with the highest population density in Africa [30]. Exposure to adverse health risks is rising with high rates of motorization and increasing levels of air pollution. In Port Louis, transport planning is reactive to the autonomous changes including population and spatial growth, with heterogeneous vehicles using limited and non-adapted road infrastructure. The central business district is car-oriented with very little green space and pedestrian movement limited to unsafe sidewalks.

The first author conducted an HIA aimed to estimate the health impacts of shifting transport modes by designing a PQHIA. This idea was proposed by the first author to different local stakeholders during previous research in 2016, which focused on the impact of urban and transport planning on citizens and their needs in Port Louis [25]. This HIA had, in addition to being policy relevant, a scholarly intention aiming at piloting a feasible model accounting for restricted resources and limited data. The quantitative component of the HIA consisted of estimating averted deaths per year and the economic value of health loss by assessing the health determinants of air pollution, traffic deaths, and physical activity for the adult population in Port Louis. The methods for the quantitative process have been published elsewhere [24]. The qualitative component of the HIA consisted of participatory methods to scope context, select health indicators, co-design HIA scenarios which are used as projections for the quantitative assessment, report results, and obtain feedback, as described below.

#### 2.1.1. Sampling and Participant Profiles

The study participants (n = 14) were recruited using purposive and snowball sampling. The sample consisted of stakeholders including laypersons, health practitioners and policymakers from different communities (see Table 1). We settled on the sample of 14 individuals based on their interest in HIA, their commitment to participate in the iterative process, and positions to potentially act on the outcomes of the HIA, as it is commonly practiced in qualitative methods for HIA [31] and also reported elsewhere [32]. The purpose of the sampling was not aimed primarily at representation, or at meeting the WHO requirement of a “large” number of stakeholders, but nonetheless special care was applied to ensure balance across communities of interest. An initial contact list was drafted with leaders and head of projects in both public and private sectors and involved in implementing different urban transport interventions and policies. Additional stakeholders were added based on recommendation from initial contacts. The government officials were chosen based on their affiliation to ministries of health or transport planning and following their availability. The citizens were contacted following events (during the sampling period) where they publicly expressed (in radio interviews or in newspaper articles) their concerns on the effects of urban transport planning on citizens. The same individuals participated in the interviews (n = 14), FGDs (n = 2) and feedback exercises (n = 6). All activities were conducted using semi-structured topic guides (see Appendix A).

#### 2.1.2. Individual Interviews

A semi-structured guide (see Appendix A) was used to elicit opinions on priority needs and challenges related to health and urban transport planning. The questions covered during the individual interviews addressed the following points:Urban transport planning (UTP) measures that stakeholders are most familiar with;Different factors stakeholders consider to lie in the interface between urban transport planning and health;Factors missing in the current situation and challenges face by the UTP sector; andIdea of a healthy, feasible and sustainable UTP system and what is needed to achieve this.

#### 2.1.3. Focus Group Discussions

The focus group discussions enabled the same stakeholders to discuss, contrast and develop their views in a group setting. During the focus groups, stakeholders were invited to discuss the following points:Their individual experiences as citizens, their needs and their priorities;Their opinions on the 3 proposed scenarios;How their visions differ and clash with the 3 scenarios; andWhether they can reach similar endpoints.

#### 2.1.4. Feedback Exercise

The feedback exercise was used to report the HIA results and as a debriefing session with the participants. Due to the coronavirus pandemic (2020), these sessions were conducted on virtual platforms (Zoom, Skype and Google meets). During the feedback exercises, the following points were covered:Reporting of baseline exposure data and final HIA results;Relevance of HIA outcomes to stakeholders’ positions and fields;Re-integration of HIA results in the society;Feedback on participatory HIA process; andReview and in some cases co-drafting, of the policy brief delivered to the authorities (see Appendix A).

#### 2.1.5. Fieldnotes

Fieldnotes were taken during recorded meetings with the study participants to collect primary data for the qualitative component of the HIA (see Table 1) and also during non-recorded meetings with external participants to collect secondary data for the quantitative component of the HIA (see Section 2.1). These stakeholders worked across five public ministries, 3 parastatal bodies and 2 private companies (see Table 2).

### 2.2. Quality Criteria Assessment

We used Lincoln et al.’s system of criteria for assessing and validating the qualitative participatory approach used to build the framework [33]. The five quality criteria include: credibility, transferability, dependability, confirmability, and authenticity. We used different proxies (questions developed by consensus with co-authors using field notes and field experience) for each quality criteria (see Table 3).

The *internal validity of participation* (credibility) was ensured by a prolonged and consecutive engagement with the 14 participants using different methods and in different locations. The attendance rate of the stakeholders to all activities was 100%. The stakeholders were able to express feedback and debrief on the process during the feedback exercises.

*External validity of participation* (transferability) was ensured by practicing knowledge exchange between the participants’ responses and researchers’ knowledge. For instance, stakeholders voiced concerns about the use of electric public transport, walkability and car overuse. These issues echoed the researchers’ knowledge of similar challenges faced in comparable LMIC settings. Stakeholders also identified health risks that are well reported in the scientific literature on HIA modelling. The subsequent meetings focused on the co-validation of scenarios and provided the opportunity for stakeholders to exchange views. The research also used articles about similar experiences to triangulate with findings from exercises with participants.

The *reliability of the approach* and *consistency in findings* (dependability) was ensured by using a collaborative process, structured topic guides (see Annex A) and recordings of sessions. The fieldwork materials were documented and indexed to ensure reliability.

*Confirmability* was ensured by gathering data at different stages of the framework and ensuring that stakeholders could be contacted for data clarification. Contact details were collected on informed consent sheets and used to confirm availability for participating in subsequent research stages.

Finally, the *integrity of participation* (authenticity) was ensured by ensuring that different views were fairly represented. Purposive sampling was used to recruit stakeholders from diverse economic backgrounds and sectors. Stakeholders were given the opportunity to add and modify their views and comments during different sessions. During the FGDs, there were different rounds of reviews after each topic to ensure that the viewpoints from different participants were contrasted, noted and considered for modifying research design and implementation (further details in [24]).

## 3. Results

We present an exploratory framework reflecting the process of PQHIA (Figure 2). The framework enables operationalizing HIA, showing how to engage stakeholders and overcome the political, economic and cultural barriers of implementation. The qualitative analysis of interviews, FGDs and feedback exercises show that costs for participation, HIA knowledge and interest, multi-sectoral coordination and multi-level stakeholder engagement influence PHIA execution. Factors such as social responsibility, policies, citizen participation and data availability influence the uptake of PHIA outcomes to have policy relevance. In our case study, good governance, resourcefulness and evidence-based policy making culture enable PQHIA execution and policy uptake in LMICs. The framework consists of three components:(1)A three-layered circular figure depicting the five main HIA stages. For each stage, a summary of participatory activities (intermediate circle) and outcomes (outer circle) is proposed.(2)A set of eight factors that can influence the process of PQHIA are presented as grey background rectangles.(3)A dotted rectangle presents three areas of opportunities for proper integration of PQHIA process in the outside environment.

### 3.1. Component 1

The framework depicts the participatory approach as a central element. The initial model (Figure 1) was modified from a linear to a circular process of the five main stages (screening, scoping, appraisal, reporting and monitoring). In our study, secondary data collected from existing monitoring databases (see Section 2.1.5) and information used during informal meetings were used to define the scope of the HIA, to map the relevant stakeholders to interview, and influenced the discussion on which indicators to examine. There was no disconnect between monitoring and screening stages. A full description of the content of the PQHIA including participatory activities and outcomes can be found in the fieldwork protocol (Appendix A) and previous publications [24].

### 3.2. Component 2

Eight different factors were found to influence PHIAs (influential factors) and the uptake of HIA outcomes in Port Louis, Mauritius: costs for participation, HIA interest and knowledge, social responsibility, HIA policies, data, citizen participation, multi-level stakeholder engagement and multisectoral coordination.

#### 3.2.1. Cost for Participation

Preliminary information shows that a lack of resources can hamper the nature and outcome of participation, leading to the inclusion/exclusion of stakeholders in participatory processes [23]. Since there are very limited data on how much participation actually costs in the broader participation literature (ibid), we used our experience from the Mauritius case study to explore these costs in more detail. Initial assumptions of participatory processes were that participants would be willing to volunteer their time and resources for such activities as the outcomes would be to their interest. As can be seen from our data, we did not cover any costs for the participants to contribute—so, our 14 participants participated entirely on a voluntary basis. This may have meant that some other stakeholders are excluded because they do not have the resources to participate both in our case study, but also more generally in LMICs.

The costs of engaging these 14 participants during an 8-month period of fieldwork amounted to 299 Euros; 7% of the entire fieldwork process for conducting the PQHIA. This excludes the plane ticket fares from the researcher’s work location (Spain), the salary of the researcher supervisors and the statistical team supporting the quantitative appraisal of fieldwork data. These costs were excluded because participatory activities on the field site in Mauritius would have still been possible without considering them. It is important to note, however, that if the municipality should undertake such an activity, the cost of the employees who organize and undertake the impact assessment will need to be added. The cost of PHIA (and HIA in general) has been identified as a major barrier to policy-level HIA [8,11,34], yet few studies have specifically reported on the actual costs of the process, leaving empirical uncertainty on the nature and extent of such barrier. This also creates uncertainty as to who is expected to fund the HIA and whether the decision to apply participatory approaches is based on financial affordances. With participatory activities covering 7% of all costs and lasting 44 days (see Table 4), our case study shows that participation in HIA does not need to be expensive or time consuming. Nonetheless, costs are influential in deciding the sample size of the population to be included in the research and the duration of the PHIA.

#### 3.2.2. Stakeholder Knowledge and Interest in HIA

Stakeholder interest in HIA is a key precondition for ensuring the effectiveness of participatory HIA process and may influence the uptake of outcomes. HIA effectiveness depends on the level to which stakeholders are (1) willing to participate, (2) knowledgeable of impact assessments, and (3) interested in, and have the authority to, integrate HIA outcomes in their work. In the case study, willingness to participate was secured by using stakeholder interest in HIA topic of HIA as a criterion for the sampling participants, following [35]. However, it was difficult to assess stakeholders’ current knowledge of HIA and their understanding of the HIA process in Port Louis. In other places, it is also still unclear as to what extent HIA processes are actually understood by health practitioners and decision-makers [36]. This may be justified by the fact that HIA has received much less policy and practice attention, compared to environmental impact assessment (EIA), for which 190 over 193 United Nations member states have signed legislation [37].

Although some participants had previous knowledge of HIA, this was not stated as a condition for them to replicate a HIA themselves or use HIA outcomes in their work. Only two participants had used HIA before (ID4 and ID10), and another had vaguely heard of HIA via EIA (ID11). In hindsight, we could have asked them about their HIA experiences in terms of the substantive and procedural aspects. One participant, a local transport provider, was interested in the visibility of issues (e.g., road deaths) that can be measured by HIA. His interest, then, was not only a precondition for participation, but also a driver to subsequently consider HIA outcomes. In the following, he comments on policy makers’ interest in transport issues: *‘I think it’s all about the visibility… of the issues […] if you say anything that will reduce road death, they [policy makers] will definitely consider it because that’s an issue* (IDI 02).

The literature confirms the importance of the ‘visibility’ of issues in the public domain [38]. Participatory impact assessment, in particular, has demonstrated how revealing ‘hidden troubles’ allows greater insights into unmet health and social needs [39]. Bringing health issues into the public domain has demonstrated that priorities for action affect physical environment, social support, and the use of epidemiological data. Participants already familiar with HIA had clear ideas on how actions should be taken and how HIA should be developed, as they were willing to use it only if it was *‘adaptable’* and *‘easy to use’* (ID01), and *‘if it help[ed]’* (ID03). Another participant highlighted that her interest in using HIA would increase if her work benefited from the external value of HIA—for instance, through the opportunity to calculate the welfare costs of health losses, especially as it is expressed in monetary terms. She commented:


*‘In Mauritius, as soon as you come with monetary values, people take you seriously (…) We carry much attention to this issue [road accidents], yet, it all comes down to money if this is to be taken seriously’*
(ID07)

Thus, when an issue become visible and a quantifiable monetary value, it may have a higher policy impact. This confirms our hypothesis that to increase stakeholders’ active involvement in PQHIA and successful uptake of HIA outcomes, it might benefit from evidence of the welfare costs of inaction versus the economic and social benefits of identifying issues and risks before their impacts are felt.

#### 3.2.3. Social Responsibility

Several participants mentioned that the (corporate) social responsibility of decision-makers is an important factor for executing PQHIA and related policy uptake. Participants perceived social responsibility as the alignment in decision-making between words and actions. A participant explained that social responsibility was lacking when political intention to change urban and transport policies had been declared but not implemented, but no action had been deployed:


*‘lately, they [policy makers] organised big consultation meetings and declared they would improve green spaces, increase pedestrianizations, plants trees on the Citadelle. All the newspapers reported this…But I see nothing, nothing, nothing...’*
(ID13)

The participant’s frustration with the lack of action by decision-makers reinforces our point on the need for PHIA to have outputs that can be realistically implemented. Similar tensions in the follow-up of good intentions expressed in public declarations have been reported in the field of impact assessments. Processes grounded in social responsibility have been proposed to encourage the systematic assessment of health impacts in local policies [40]. Thus, social responsibility becomes connected to the idea of ‘accountability’ of decision-makers towards their promises: *‘If one makes a mistake, it is responsible and necessary to correct them’* (M, FGD2).

Networks such as the WHO’s Health Cities network and the Milan Declaration have suggested that a way to build accountability is via ‘making health and environment impact assessment part of urban planning decisions, policies and programmes’ [41]. In Port Louis, stakeholders reported that social responsibility would increase if policy makers adopted gradual changes, ‘*retro-fitting*’ (ID 03) rather than proposing new development plans at every round of decision-making. This would increase ‘*pre-visibility*’, ‘*resolution*’ and *‘follow-up’* of plans (K, FGD1). Without this continuous, accountable and reliable process, the planning strategy has no value: *‘there needs to be follow-up in the decisions taken’* (M, FGD2).

Finally, social responsibility is closely related to the ability of policy makers to go beyond political mandates and balance short-term needs with long-term benefits (IDI10). In this sense, social responsibility is evident in the *‘difference between the political figure and the statesman the political figure thinks about policies for the next elections, the state figure thinks about policies for the next generations’* (IDI14).

#### 3.2.4. HIA Policies

Participants reported that policies and legal frameworks are necessary to encourage the use of tools such as HIAs. One participant commented:


*‘The underlying issue lies in the lack of regulatory policy frameworks: a legal administrative protocol to use tools such as HIA and a framework that goes beyond government or private mandates’*
(R, FGD2)

Participants also confirmed that they would use and conduct HIA only if it were regulated by law or imposed on top of their required workload (ID2, ID5). One participant says: *‘If the regulations do not force or regulate us to use HIA, then we will not use it’* (ID 2). Although few LMICs have established HIA legislation and no African country regulates HIA [42,43], large multilateral organizations such as the African Union have recognized the need for developing HIA policy in developing countries [44]. Indeed, developing countries showing strong political commitment to HIA will not only raise awareness of its use, but will effectively contribute to building technical capacity for HIA [45].

Past experience with EIA shows that policy makers struggle to satisfy policy goals including political, economic and social aims, are challenged to find ways for enforcing EIA recommendations, and that governments are concerned about how to integrate different policy initiatives [46,47]. Stakeholders from Port Louis highlight similar issues. For example, even though governments have been talking of electric cars for 20 years, based on EIA reports, no progress has been made (B, FGD2).

In response, they suggest that policies for planning should be applied long term especially those that focus on the deployment of impact assessments:

*‘If we have policies, we need to apply them sustainably’* (R, FGD2)*. Finally, ‘having law is not enough, we need to enforce them’* (IDI4). Finally, a participant stresses that there should be continuity between policy-making and policy integration in order to translate HIA outcomes into actions. *‘It should be the same policymakers who formulate [policies] and integrate [policy-driven actions and interventions] too’*(ID07)

#### 3.2.5. Data Availability

Participants reported that data availability provides the opportunity to assess a larger scale of factors impacting health. One of them remarks *‘how can we predict health impacts effectively if we do not have baseline data?’* (P, FGD2). Lack of data on air pollution could be compensated by data on the fuel used by public transport (ID2). Setting up epidemiological databases could address the issue of data scarcity (ID 10). Beyond a lack of data, various concerns were raised about how existing data are collected, managed and used:

*‘I think they do not use what they have intelligently, they do not know what they have) […] there is a lot of data management equipment that is being under-utilized’* (ID 10). Such data (e.g., that there is better air quality on Sunday because people do not travel as much) should be directly used to encourage and increase awareness of the public(ID12)

Adequate data collection and management is useful to examine how environmental factors affect population groups differentially. Specific data are needed to better assess the health impacts of transport on those most exposed, such as traffic policemen and slum dwellers; and whether people actually use green spaces in a city (ID12). Understanding who is benefitting the most, or who is most impacted by policies, is critical (S, FGD1).

Data (quantitative or scientific but qualitative or contextual) can also serve to inform policies and avoid mismatches between what people are doing and what policies are regulating. For example, data on travel mode shares and travel behaviors is needed to design car reduction strategies (R, FGD2). Lack of adequate data feeds misinformation and misconceptions, which in turn makes the adoption of novel strategies for health harder, as in the case of cycling, which is perceived to be risky:


*‘People are concerned about their security on bicycle, but this is a lie that protects the car […] of course the road is dangerous, especially for the two-wheelers, but this should not be encouraging the restriction of cycling modes and favour the use of cars’*
(G, FGD1)

Finally, obtaining and using data also means that policy makers can explore and use other types of evidence from similar settings and countries (particularly, in the region of Africa) when they consider designing policies (IDI02).

#### 3.2.6. Multi-Sectoral Coordination

Following stakeholder opinions, multi-sectoral coordination prompted by PQHIA is essential to (1) interact with agents from other fields, (2) to use data and evidence from other sectors, and to (3) tackle different issues with the same strategy. Within our sample, multi-sectoral coordination was already happening prior to HIA and was fostered during HIA. For instance, a public service urban planner was working with the head of project of an international multilateral agency and at the same time collaborating with a consultant from the private sector (ID02). Furthermore, one participant suggested that different forms of evidence (other than travel data) are needed to regulate public transport in Port Louis (ID02). Using multi-sectorial approaches enables one to tackle different issues and how they intertwine. Transport-related health issues and national security influence each other (ID12).

Participants argued that a lack of cross-sectoral work leads to poor coordination of actions and policies. One participant suggested that an official board should be created specifically for overviewing projects that demand a cross-sectoral approach and coordination between ministries and different stakeholders (ID 11). This could build on experiences in other sectors such as on employment protocols, occupational health and immigration laws (ID 07).

Not adopting cross-sectoral approaches can lead to the design of sector-specific policies that address the wrong problems. For instance, participant (ID 09, Z FGD 1) mentions that her choice of travel mode is not based on health benefits per sé but on the level of physical safety: *‘I will not choose to walk, even if it is better for my health, […] because of rapists potentially hiding’* (Z, FGD1). Therefore, even if policies to increase walking are designed, they are not efficient if they overlook personal safety issues. Participants called for addressing health in different policies and sectors (e.g., agriculture, trade): *‘sustainable transport is linked to a lot of different things, agriculture, oceans, partnerships, income trade, you name it, it is all linked*’ (ID09). Additionally, such an approach enhances the link between sectors, for instance between transport and use of natural land and urgency of protecting resources (J, FGD2).

#### 3.2.7. Multi-Level Stakeholder Engagement

Engaging stakeholders from different levels of decision making within one sector of interest was recommended (ID07) as it can provide contrasting perspectives and facilitate priority setting of issues. However, the gaps between different actors and levels of decision-making action in the health sector might hinder collaboration:


*‘Half of the doctors are very interested [in impacts of environment on health], but there are no avenues for them to actually get data […] All the policies or everything that they produce make sense but the human reality is different...it would be better if they were aware of the meaning of their work’*
(IDI 10)

Multi-stakeholder collaboration intertwines with other issues considered above, such as the importance of resources, data, policies, and multi-sectoral coordination, and how all come into play in making HIA possible.

#### 3.2.8. Citizen Participation

Within PQHIA, participants considered that a space should be provided for citizens to contribute to more inclusive processes of policymaking: *‘I believe that citizens should be responsible and accountable enough to ask what they need and want’* (ID 12). PQHIAs reflect the willingness of HIA practitioners and stakeholder to consider citizen’s views and lay knowledge alongside expert opinions and scientific data. The process of triangulation implied in combining several types of evidence and from different sources increases the value of the process. In some contexts, decision-makers may value information sourced from communities; citizen expectations can help them decide on priority issues (IDI 13). Therefore, PQHIA can provide channels for such evidence to be rapidly conveyed to them, in a clear and transparent manner. This is also valid to feedback HIA findings to the community, to strengthen equity and gather baseline information for routine impact assessment. The literature reports that citizen participation is a complex process that can be expensive [23] and requires citizen organization to have effective influence [48]. In developing countries, however, not enough evidence exists to verify such statement and our case study reports otherwise (see previous section).

### 3.3. Component 3

Three areas of opportunities were found to facilitate implementation for HIA: good governance (political), evidence-based policymaking (cultural) and resources (socio-economic). They are represented in a dotted rectangle delimiting PQHIA with the outside environment (see Figure 2). Without investment in these three areas of opportunities, PQHIA faces the risk of not being properly integrated, limiting the ways in which countries and institutions can use and benefit from the tool.

#### 3.3.1. Good Governance

By facilitating multi-sectoral and multi-level coordination, PQHIA provides a point of focus on good governance, especially in countries with limited resources. Governance is the manner in which power is exercised in the management of a country’s economic and social resources for development to be able to properly execute governance, government integration and internal collaboration is a key factor [49]. Collaboration between departments (multisectoral) and between levels of government (multi-level) can provide a more effective identification of needs and gaps, data collection, policy elaboration, evidence integration and comprehensive policy execution. In this sense, governance structures aimed to promote collaboration makes any HIA proposal and results be integrated in the system to be effective. So, how HIA demands and outcomes/outputs are integrated in real policy also depend on how these multi-sectoral/level structures are emplaced and working. HIA requires the leadership of the health sector but most of the policies require other sectors and levels of authority to be executed. The lack of good governance will diminish HIA quality and utility. Therefore, PQHIA in a context of good governance can be optimized as a tool to provide evidence, use robust data, maximize health, minimize risks, and integrate recommendations. HIA can also help to identify governance gaps and reinforce the need for good governance structures to support evidence-based interventions.

#### 3.3.2. Evidence-Based Policymaking

The primary output of HIA is scientific evidence that can inform decision makers early in the process of policy development and ensure that health impacts are not overlooked. Validating evidence-based policy making as an area of opportunity for HIA practice provides scope for increasing political authority and diffusing impact of HIA outcomes. During an interview, a public servant working at the Ministry of Health reported that outcomes resulting from impact assessments are taken seriously in Port Louis (ID 04). There are environmental impact committees evaluating health components and providing recommendations. Yet, the participant reports that ‘*legislation can be enough [to promote use of evidence] but enforcement is a problem.’ (ID 04).*

Different public health frameworks facilitate and promote the practical application of scientific evidence derived from evaluation tools such as HIA. For instance, the Health in All Policies (HIAPs) approach is widely applied to support evidence-based public policy and benefit local decision-making processes [26,50]. However, in LIMCs, such frameworks are weak, making it complicated to assess whether or not outcomes are effective for evidence-based policy making. In the case study, a policy brief was drafted and submitted to decision makers after the HIA was conducted (Appendix A). Various participants agreed to co-author on the policy brief to show their interest not only to support HIA practice but also to enforce its impact. On a wider scale, however, securing evidence-base policy making practices may imply building resources and skills, planning for effective engagement strategies for key stakeholders, defining evidence-based monitoring and evaluation mechanisms for health data, etc.

#### 3.3.3. Resources

The need for human and economic resources that enable decision makers to consider health impact outcomes emerged various times in the participatory study. While discussions about human resources traditionally evolve around distribution of workforce and manpower, in the context of PQHIA, human resources were expressed in terms of the ability, attitude and skills of decision makers to analyze, understand and predict the health impacts their strategies and interventions that would compel them to consider HIA outcomes and strategically plan in advance (ID 12). Yet, addressing human resources (skills and attitude) without the support of economic resources is limitative. Decision makers need both ‘intentions’ (see social responsibility) but also the means to follow these intentions (e.g., skills, economic resources). Elsewhere, this has been referred to as developing resourcefulness; i.e., ensuring access to economic, civic and intellectual resources at the same time [51].

Economic resources are necessary to address environmental and health concerns. For example, one participant commented that *‘A country focuses on health and traffic safety only if a given milestone towards development has already been reached’* (R, FGD2). His statement, which effectively mentions the level of ‘economic development’ as a precondition for implementing PQHIA, speaks directly to the academic debate on the links between GDP and environmental policies. Different studies have observed that countries tend to react to increasing levels of environmental damage only after a certain level of pollution (and health impacts) and GDP is reached, as expressed in the so-called ‘Environmental Kuznets Inverted U’ curve [52]. However, the question is whether the idea that one needs to develop first before considering the environment and health issues is an increasingly problematic one, because inappropriate investments (or access to economic resources) cannot easily be phased out before their useful life-time without being extremely expensive (e.g., a city that has heavily invested in fossil fuel using private transport cannot easily shift into a city that has electric public transport and lots of bicycle paths without creating stranded assets).

Another participant raised the complexity in defining what this threshold of economic development may be, because, in his view, when health is considered in non-health sectors, this is often unplanned and unexpected:


*‘In Mauritius, the best urban and transport planning achievements, including those that benefit health, have been completed without specific strategy and clear planning. There is local pride in this approach to development, leaving very little space for critique’*
(IDI08)

While it is possible to frame development through GDP levels (as in the previous paragraph), this participant shows that there may exist different ideas about urban development or progress, and how they interrelate to the possibility for HIA to bring about change. Such ideas may pertain to dimensions beyond economic resources and into human resources as mentioned previously. They can also be linked to local knowledge, institutional cultures towards both strategic (or in this case, unstrategic) planning, further supporting our point that the current framework may need to be adapted across contexts.

## 4. Conclusions

The proposed PQHIA framework is a tool for guiding participation in quantitative HIA while overcoming political and cultural barriers for implementation. It was developed based on fieldwork conducted in Mauritius and can be considered for wider applicability. The findings are valuable given the scarcity of HIAs in LMICs. In a systematic review of 57 HIAs in LMICs, only seven studies reported participatory approaches [19]. They showed how participatory HIAs assist to set assessment boundaries [53], to clarify expectations of HIA practitioners, to disaggregate different determinants of health [54] and to promote collaboration with HIA practitioners from other countries [55], but none explained in detail and with precision how the participatory process should be operationalized. The aforementioned studies were also participatory qualitative, and not quantitative, HIAs.

The framework brings additional evidence to previous models addressing process [56,57] and HIA implications for policy-makers in LMICs [58]. It stands out, however, by reporting different factors underlying PQHIA at every stage, using empirical data and recording stakeholder insights before, during, and after the appraisal stage. The framework fills existing gaps in the research and complements studies that do not adequately report the theoretical and practical process leading to specific methods for participation [15]. It also provides an insight on how to tackle the issue of the mismatch between how participation is described and perceived and the actual realities of involved [13]. By including a diversity of stakeholders in the process the framework caters for different perspectives in the participatory HIA process [59]. Finally, the engagement of stakeholders in the Mauritian case and the resulting framework is promising and contrasts with findings from a richer setting that stakeholders perceive participation as a burden and a constraint characterizing HIA [60].

The proposed framework reflects how contextual factors can influence participatory HIA. Wismar el al. 2007 showed that the effects of HIAs on decision-making processes are complex and can vary significantly: (a) they directly affect the decision being made; (b) they do not affect the decision but raise awareness of health issues; (c) they have little impact because the decision was already favorable to health; and (d) they are ignored or dismissed by the decision makers [61]. Even if this was possible, claiming that a standardized one-size-fit-all framework can encapsulate all changes attributable to the practice of participation in HIA would require a large number of empirical studies and an acute understanding of PHIA implementation in settings such as Mauritius, both of which are currently lacking. Therefore, we propose our framework as a guiding tool rather than a standardized framework for participatory HIA. In the process of building this framework, some lessons have been learnt:Methods for participation (e.g., activities, sample size and study period) should be carefully planned, in advance, based on budget and time available.The flexibility afforded by choosing the type, form, and duration of HIA alongside local communities is crucial for stakeholders to use and most benefit from PQHIA.Focusing on the areas of opportunities highlighted in the framework can have wider benefits on governance systems, policy-making practices, and access to resources in LMICs.

Several challenges remain and should be further studied. The study presents results based on a small-scale sample (14 participants). Even if large samples are seldom used in the context of participatory impact studies of transport and there is a consensus that cost affects sample size and time of study [62], implications of how to scale methods to national level or in another context need to be examined carefully. Small sample sizes and their limitations to extrapolate findings are a common challenge in the field of impact assessments [32]. Studies reporting stakeholder involvement were all in OECD countries covering between 14 and 52 participants [63,64,65,66]. Those involving more than 100 participants were rare and conducted in rich countries with the most history and experience in HIA, such as Canada and Switzerland [67,68].

In this LMIC-based case study, the choice to work on a small scale was deliberate and adapted to satisfy scholarly objectives. The samples needed to be manageable to ensure long-term engagement from stakeholders across different meetings and events, the timeline was set, the resources and costs were limited and the HIA was led by one person (main author) who did not reside in the country of study. The selection of participants based on their interest in HIA carried the risk of sampling bias that was reduced with careful research design and transparent sampling frame. The sample also consisted of participants who were able to participate without financial stipend. This may not be afforded in other situations or settings. Sampling participants for qualitative and participatory research phases requires a judgement of who is irrelevant and where to draw the line, but also to consider how different views are included in the final decisions and the way in which they are implemented.

It is possible that slow institutionalization of HIA in LMICs (at national level) increases the complexity of incorporating stakeholders with environment and health backgrounds [32]. Yet, we believe that the sample satisfied the focus and limits of this study in terms of city-level approach, the intensity of their engagement (recurrence and thoroughness) and inclusion process in conducting the HIA but also disseminating its results. This study supports current evidence that stakeholder engagement is an efficient method to improve the quality and relevance of HIA [35,69,70]. Even if some concern may remain on what ‘minimum level’ of participation is required for a quantitative HIA to be defined as PQHIA, the quality criteria assessment we applied (see Section 2.2) was helpful in exposing the framework’s ability and potential to guide planning and implementation. In addition to the formal participatory activities with 14 participants, a large number of informal meetings and fieldnotes (see Section 2.1.5) were collected during the PQHIA. Ensuring the robustness of qualitative data using criteria is invaluable in ensuring scientific rigor, researcher reflexivity, and ethics towards participants.

To conclude, our study recommends the use of PQHIA to bring health to non-health sector agendas such as urban transport planning. By allowing for the greater dissemination of HIA outcomes, PQHIA raises health awareness among wider members of the community. PQHIA engages stakeholders at every stage of the process, increasing their knowledge gradually and providing various points of entry for HIA to impact their individual environment and sectors. Finally, PQHIA responds to an urgent need to combine increased knowledge on deteriorating health determinants and experiences with impact assessment as a potential solution to safeguard the health of people and the planet.

## Figures and Tables

**Figure 1 ijerph-17-07688-f001:**
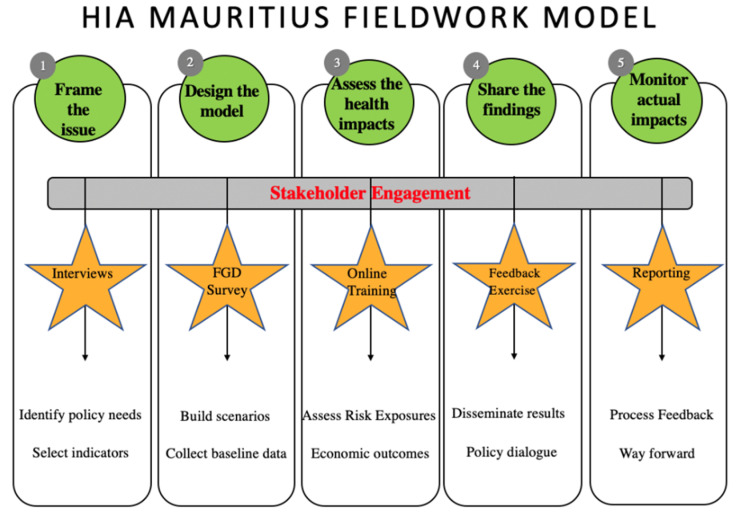
Baseline case-study fieldwork model.

**Figure 2 ijerph-17-07688-f002:**
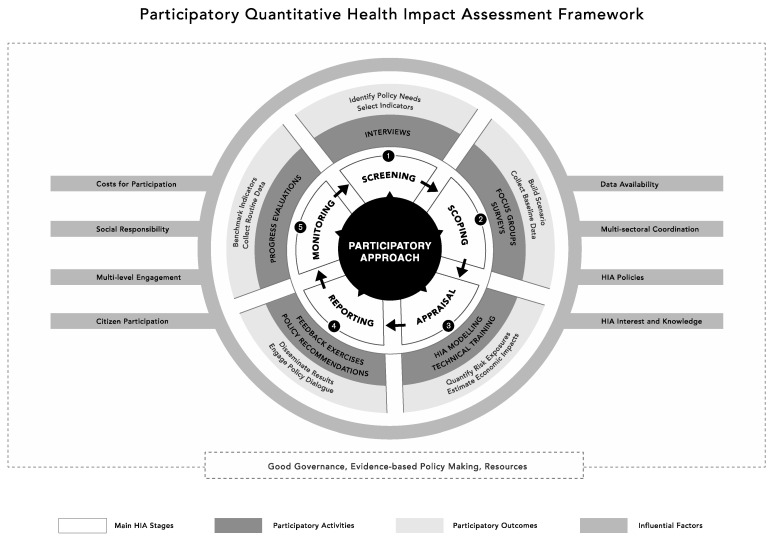
Participatory quantitative health impact assessment (PQHIA) framework. Note: for sake of visual clarity, the recommendation stage is presented as a sub-activity of the reporting stage, even if in practice this holds its importance as a stand-alone stage.

**Table 1 ijerph-17-07688-t001:** Participant profiles.

Communities of Interest	Expertise (Information Held)	Reason to for Inclusion
Community-based organization	Expert (Ecosystems)	Active role in liaising between communities and developers of private urban project
Service provider/industry	Consultant (Sustainable development)	Consults for public and multilateral organizations on the environmental impacts of land and sea infrastructure projects
Elected official	Adviser (Land Transport)	Provides expert advice and strategies to the high-level politicians on transport related policies
Elected official	Permanent secretary (Medicine and health)	Reviews environmental impact assessment reports on national projects in order to identify health risks
Industry	Planner (Urban planning)	Leads on private and public transport urban planning projects such as main bus terminal
Public agency	Statistician (Land transport)	Updates and monitors data on land transport such as traffic incidents and deaths
International multilateral organization	Head of department (Sustainable development)	Reports on the advancement of sustainable development targets on the island including SDG 11
Public agency	Head of department (Traffic Planning)	In charge of deploying public transport strategies and involved in the new light-rail public transit system
Public agency at parastatal level	Technician (Sustainable economic development)	Works on establishing urban standard guidelines for economic development focusing on investments in transport, social housing and real estate projects
Public Agency at municipal level	Municipal agent (Town planning and services)	Works at municipal level on housing and transport initiatives and municipal policies
Industry	Executive and board director (Economic investments, Food services and Sustainable development)	Directs decisions for different companies focused on services and investments in the city of Port Louis
Resident	Journalist (Urban development)	Critically analyses and reports on urban development projects across the island
Resident	Politician (Social and economic development)	Leads a stand-alone political party with expertise in sustainable economies
Resident	Social worker (Health and social justice)	Provides support and leads initiatives supporting the urban poor in the capital

Source: [24].

**Table 2 ijerph-17-07688-t002:** Non-recorded meetings for secondary data.

Meeting with	Data Needed	Category	Institutional Affiliation
Health statistician	Population Census by gender and age	Demographic	National Institute of Statistics
Land transport agent	Mode and time of travel by gender and age	Transport	National Transport Agency Ministry of public infrastructure and land transport Private transport provider
Environmental expert	Air pollution emissions inventory and database	Air pollution	National Environmental Laboratory Ministry of Environment
Public Health statistician NCD expert Permanent secretary	Health and Physical activity survey data by gender and age	Physical activity	Ministry of Health
Police officers Hospital staff Transport expert	Records of traffic collisions Vital registration statistics Hospital records	Road deaths	Police headquarters and traffic offices Transport agency National institute of statistics Hospitals
Health researcher Health statistician	Vital registration statistics Burden of disease data	Burden of disease	National institute of statistics Health agency University of Mauritius
Climatologist Weather expert	Daily mean temperature	Heat	Mauritius Meteorological station
Urban planner GIS expert Permanent secretary Architect	Map of land use Topography layers Public transport route maps	Land use	Ministry of Housing and Lands Economic Development Board Ministry of Local Government and disaster risk management

**Table 3 ijerph-17-07688-t003:** Description of quality criteria.

Criteria	Description of Criteria	Proxy
Credibility	*Internal validity of participation*: Do participants feel that the findings represent their experience?	Was there a prolonged engagement with participants? Was there a debriefing session with the participants?
Transferability	*External validity of participation*: are the findings applicable to other contexts?	Are participants’ responses in harmony with researcher’s experience? Is there scope to provide a detailed description from both sending and receiving ends?
Dependability	*Reliability in participation*: are the findings consistent?	Can the researcher use documents and methods to check if research strategies have effect?
Confirmability	Can we confirm the findings using data analysis?	Can the findings be confirmed if data are recollected and analyzed?
Authenticity	*Integrity of participation*: are all the different views fairly represented? Did the process stimulate action from participants?	Were viewpoints from different participants considered? Did the participatory process lead to participants acting on HIA outcomes?

**Table 4 ijerph-17-07688-t004:** Cost and working days.

Activity	Item	Number of Items	Price Per Item	Total MUR	Total EUR	Working Days
Survey Data Collection	Fieldworker (FW) salary	8	12,000 rps/FW	96,000	2412	20
	Bus trips for FW	320	25 rps/trip	8000	201	
	Weekend bonus	8	1000 rps/wkend	8000	201	4
	Software and hosting	One-off fee	24,000 rps	24,000	603	
Intern Support	Intern support (1)	20 hrs	20 hrs	1875	47 *	4
	Intern support (2)	20 hrs	20 hrs	1500	38 *	4
FGD	Room location	2hrs	750	1500	38 *	1
	Facilitation strategy	3 hrs	1000	3000	75 *	1
Secondary data	Heat data	5	200 rps	1000	25	10
	Cartography Layers	4	4000	16,000	402	
	Transport	40 hrs	100 rps/hr	4000	101 *	
	Outline Planning Scheme	1		5000	126	
				Total	4268 EUR	44 days

* Costs related to engaging the 14 participants only.

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
