# Peer review of "Framework for Participatory Quantitative Health Impact Assessment in Low- and Middle-Income Countries"

_ijerph, 2020, doi:10.3390/ijerph17207688_

Round 1
Reviewer 1 Report
Methodology is fair described. However, balancing the need to represents stakeholder opinions and article length is unachieved. Maybe reconsider some direct citations from stakeholders if they are unnecessary. I moslty appreciated the discussion points and the considerations about strenghts and limitations of the approach utilization.
I provided notes at lines 267 and 270 pag. 8, within the attached document
note 1: referring to an "external" framework looks redundant to explain the case study approach. Here it partly modifies the basic framework and is well described.
note 2: this is unclear. please provide one more comment about using this order. the usually recognized framework describes screening and then scoping.

Author Response
Dear Editor,
Many thanks for the very constructive comments you have sent us. In essence, the reviewers have three main comments: (a) shorten the article length (b) review the steps in the main framework and (c) embed arguments about sampling size in HIA.
Our detailed responses to the individual points are provided below in the table.
|
|
Reviewers comments 1 |
Our response |
|
1 |
Methodology is fair described. However, balancing the need to represents stakeholder opinions and article length is unachieved. 1. Maybe reconsider some direct citations from stakeholders if they are unnecessary 2. I mostly appreciated the discussion points and the considerations about strengths and limitations of the approach utilization. |
Thank you.
1.1 The article has been shortened and some direct citations removed. 1.2 Thank you |
|
|
I provided notes at lines 267 and 270 pg. 8, within the attached document note 1: referring to an "external" framework looks redundant to explain the case study approach. Here it partly modifies the basic framework and is well described. note 2: this is unclear. please provide one more comment about using this order. the usually recognized framework describes screening and then scoping. |
Note 1: The major difference with the proposed PQHIA framework (vs. the case study approach) lies in the circular rather than linear process and participatory activities depicted as a central component rather than a side-benefit of PQHIA.
Note 2: There was a mistake in the steps. It has been modified. |

Reviewer 2 Report
I have enjoyed reading this work because it deals with several topics in which I am very interested and dedicated a lot of effort. Although with less success than the authors. In general, it is a very interesting job. It deserves to be published but there are some details that I would like to draw attention to.
Abstract
Line 51- the authors affirm that “…participation in HIA does not need to be expensive or time consuming for the assessor or the participant.” However it is not clearly stated in the body of the text.
Introduction
Line 69 – Why you say that “…literature demonstrates that voluntary HIAs are more likely to influence decision-making when the process encourages participation”? Why do you exclude mandatory HIAs?
Methods
Major criticism may relay on the sampling size. The number of participants (14) can be small and it may be interpreted as a weakness ... I myself have been raised this criticism on occasions. It would be convenient to discuss the sampling size. In this sense, some arguments about sampling size in HIA and EIA studies can be found in Domínguez-Ares et al. (2020). Perception survey on the relevance of main categories of health determinants for conducting health impact assessment. Environmental Impact Assessment Review, 85, 106445.
Fig. 2.
It is not clear to me that in this scheme the scoping stage is placed before the screening stage.
Screening is always first stage in both EIA and HIA. For instance, scoping is the stage to identify whether a project/proposal needs an EIA or HIA.
Following this consultation stage, it is necessary to determine the extent of the appraisal (details of information which are required). It would be the next stage: the scoping stage.
Therefore, these stages cannot be altered.
I think this model should be revised. Otherwise, the stages' order should be clearly justified. Why to define a scope before knowing whether it is necessary to do a work or not?
Section 3.2.1 cost of participation
This part is very interesting. I congratulate the author which are able to explicitly estimate monetary costs linked to the participation process. However, it is not clear to me why some costs are excluded (Lines 294-298). Can the work be done without them?
It should be justified or included. If costs are underestimated, it could result unrealistic and reinforce the argument of those who maintain that participation is expensive (this is the opposite of what is said in the abstract of the article).
Table of costs: I find it would be also very interesting to report costs in terms of working days covered by a full-time working person at each responsibility of the participatory process.
Line 327 – please revise this value, EIA together with SEA are tools used by most countries around the World
Line 525-526 - authors say that public participation can be expensive. Is this affirmation based on the costs calculated in this study case? If yes, it should be clarify. Otherwise, it seems contradictory of what is said in the abstract of the article (again).
In general, the reading gets a bit long from lines 307 to 620. The authors should consider the possibility of cutting out some less relevant contents in this section.
Finally, let me ask the authors a question. Do you consider that costs linked to HIA stages can justify its integration into other impact assessment procedures which may already exists in Mauritius (e.g. EIA)? Please see Iglesias-Merchan, C., & Domínguez-Ares, E. (2020). Challenges to integrate health impact assessment into environmental assessment procedures: the pending debate. Impact Assessment and Project Appraisal, 1-9.
Author Response
|
Reviewers comments 2 |
Our response |
|
I have enjoyed reading this work because it deals with several topics in which I am very interested and dedicated a lot of effort. Although with less success than the authors. In general, it is a very interesting job. It deserves to be published but there are some details that I would like to draw attention to.
|
Thank you very much. |
|
Introduction Line 69 – Why you say that “…literature demonstrates that voluntary HIAs are more likely to influence decision-making when the process encourages participation”? Why do you exclude mandatory HIAs?
|
‘Voluntary’ has been removed. Literature addresses HIA as a general term. |
|
Methods Major criticism may relay on the sampling size. The number of participants (14) can be small and it may be interpreted as a weakness ... I myself have been raised this criticism on occasions. It would be convenient to discuss the sampling size. In this sense, some arguments about sampling size in HIA and EIA studies can be found in Domínguez-Ares et al. (2020). Perception survey on the relevance of main categories of health determinants for conducting health impact assessment. Environmental Impact Assessment Review, 85, 106445.
|
The number of participants is always critical. We chose a smaller number, because we wanted people to be committed to an iterative process of participation in our workshops. This is also an experiment within a PhD project. If this is to be applied by the municipal government, this would possibly require more participants.
Modified in text: see line 162 and line 665 to 687 |
|
Fig. 2. It is not clear to me that in this scheme the scoping stage is placed before the screening stage. Screening is always first stage in both EIA and HIA. For instance, scoping is the stage to identify whether a project/proposal needs an EIA or HIA. Following this consultation stage, it is necessary to determine the extent of the appraisal (details of information which are required). It would be the next stage: the scoping stage. Therefore, these stages cannot be altered. I think this model should be revised. Otherwise, the stages' order should be clearly justified. Why to define a scope before knowing whether it is necessary to do a work or not?
|
Apologies, there was a mistake in the stages. This has now been corrected. |
|
Section 3.2.1 cost of participation This part is very interesting. I congratulate the author which are able to explicitly estimate monetary costs linked to the participation process. However, it is not clear to me why some costs are excluded (Lines 294-298). Can the work be done without them? It should be justified or included. If costs are underestimated, it could result unrealistic and reinforce the argument of those who maintain that participation is expensive (this is the opposite of what is said in the abstract of the article).
|
These costs were excluded because they did not involve activities directly involved in fieldwork on the case study site in Port Louis (see line 297). The participatory components of the HIA could have been done without these costs. It could be argued that the airfare costs of the researcher who led participatory activities should have been included, but these costs are not relevant if HIA is locally led and justified. It is important to note also, that if the municipality would undertake such an activity, the cost of the employees who organize and undertake the impact assessment will need to be added. |
|
Table of costs: I find it would be also very interesting to report costs in terms of working days covered by a full-time working person at each responsibility of the participatory process.
|
These have been added. |
|
Line 327 – please revise this value, EIA together with SEA are tools used by most countries around the World
|
Revised. |
|
Line 525-526 - authors say that public participation can be expensive. Is this affirmation based on the costs calculated in this study case? If yes, it should be clarify. Otherwise, it seems contradictory of what is said in the abstract of the article (again).
|
Clarification provided in text. |
|
In general, the reading gets a bit long from lines 307 to 620. The authors should consider the possibility of cutting out some less relevant contents in this section.
|
Yes, the length has been reduced.
The additional proposed references have now been included in the paper. |
|
Finally, let me ask the authors a question. Do you consider that costs linked to HIA stages can justify its integration into other impact assessment procedures which may already exists in Mauritius (e.g. EIA)? Please see Iglesias-Merchan, C., & Domínguez-Ares, E. (2020). Challenges to integrate health impact assessment into environmental assessment procedures: the pending debate. Impact Assessment and Project Appraisal, 1-9.
|
In a publication entitled: Health Impact Assessment Legislation in Developing Countries: A Path to Sustainable Development? (2020), we argue that regulatory HIAs (HIA integrated with EIAs) provide several opportunities for developing countries. HIAs when considered as an operational tool to achieve wider agendas such as the SDGs or HiAPs can have various can bring much value and benefit from the policy landscapes set by EIAs. We state that EIA statutes in LMICs can provide procedural rules and legal levers for HIA practice. Decision makers can access information through a single process. EIA and HIA integration can involve stronger and more pragmatic collaboration between health and environmental agencies. |

Round 2
Reviewer 2 Report
Most of my comments have been correctly attended.
I think the article can be accepted in its present form.